# Stability Study of Cannabidiol in the Form of Solid Powder and Sunflower Oil Solution

**DOI:** 10.3390/pharmaceutics13030412

**Published:** 2021-03-19

**Authors:** Ema Kosović, David Sýkora, Martin Kuchař

**Affiliations:** 1Institute of Chemical Process Fundamentals of CAS v.v.i., Rozvojová 135, 16502 Prague, Czech Republic; kosovice@vscht.cz; 2Department of Analytical Chemistry, University of Chemistry and Technology Prague, Technická 5, 16628 Prague, Czech Republic; sykorad@vscht.cz; 3Forensic Laboratory of Biologically Active Substances, Department of Chemistry of Natural Compounds, University of Chemistry and Technology Prague, Technická 5, 16628 Prague, Czech Republic

**Keywords:** cannabidiol, degradation, oil matrix, stability study, tetrahydrocannabinol, cannabinol

## Abstract

Stability studies represent an essential component of pharmaceutical development, enabling critical evaluation of the therapeutic potential of an active pharmaceutical ingredient (API) or a final pharmaceutical product under the influence of various environmental factors. The aim of the present study was to investigate the chemical stability of cannabidiol (CBD) in the form of a solid powder (hereinafter referred to as CBD powder) and also dissolved in sunflower oil. We performed stress studies in accordance with the International Conference on Harmonization (ICH) guidelines, where 5 mg of marketed CBD in the form of a solid powder and in form of oil solution were exposed for 7 and 14, 30, 60, 90, 180, 270, and 365 days to precisely defined temperature and humidity conditions, 25 °C ± 2 °C/60% RH ± 5% and 40 °C ± 2 °C/75% RH ± 5% in both open and closed vials in the dark. CBD powder was significantly more stable than CBD in oil solution. Such finding is important because CBD is often administered dissolved in oil matrix in practice due to very good bioavailability. Thus, the knowledge on admissible shelf time is of paramount importance.

## 1. Introduction

It has been known for thousands years that the cannabis plant interacts with the human body on many levels: to relieve neuropathic pain [1], to lower intraocular pressure [2], to increase appetite [3], and to decrease nausea and vomiting [4]. The substances responsible for these effects are collectively known as cannabinoids, among which the most prominent are Δ^9^-tetrahydrocannabinol (THC), cannabidiol (CBD), and cannabinol (CBN) (Figure 1). Due to the fact that a combination of psychoactive THC with non-psychoactive cannabinoids shows a higher activity than THC alone, a lot of effort and resources have been invested in studying and isolating non-psychoactive alkaloids and their application in medicine [5].

CBD is the second most prevalent active ingredient of cannabis, which does not possess any psychoactive effects. According to the report from World Health Organization, to-date, there is no evidence of public health related problems associated with the use of pure CBD [6]. Only a few publications can be found in the literature concerning storage conditions and degradation of cannabinoids in general, where authors clearly specify the storage conditions and individual forms of cannabis (oil, resin, etc.) [7,8]. The mentioned papers emphasized a promising medical potential of CBD. Many clinical trials and government approvals have been completed or are currently underway. One of these trials provided lung cancer treatment to patients with adenocarcinoma [9]. Earlier works have shown that CBD has probably anti-neoplastic properties, but more recent data have indicated that CBD may also provide a significant response in patients with this kind of cancer. However, on the other hand, there are some indications that CBD oil accelerates certain types of cancers; therefore, cancer should be treated with great care and distinction as to what type of cancer is being treated [10]. Due to its promising results in early trials, there is an increasing interest in research of CBD as a dietary supplement with potential medicinal benefits.

To assure an adequate use of CBD in medicine, it is necessary to know its physical, chemical, and biological properties, and information regarding stability and shelf-life. Lindholst [11] described cannabis resin and extract as very susceptible to degradation caused by light, temperature, and oxygen, while Carbone et al. [12] provided an almost complete review of THC degradation products. A similar degradation mechanism was proposed by Layton et al. [13], who focused more on the identification of degradation products caused by the above-mentioned factors, but the results are not fully pharmaceutically acceptable due to the used methanolic matrices and length of the study [14].

The aim of our work was to investigate the chemical stability of commercially marketed formulations of CBD forms under defined International Conference on Harmonization (ICH) guideline requirements. The stability of CBD was evaluated in the form of a solid powder and an oil solution for a one-year time period, under ICH stability conditions. The study was adapted to the climate conditions of the country where the experiment was conducted, i.e., the Czech Republic. High-performance liquid chromatography (HPLC) in combination with ultraviolet-visible (UV-Vis) spectroscopic and mass spectrometric (MS) detection was used as a suitable method for the stability measurements. To the best of our knowledge, no other long-term stability study of CBD in both the above-mentioned forms has been performed to-date.

## 2. Materials and Methods

### 2.1. Chemicals and Reagents

CBD with a purity of 99.9% was purchased from Lipomed AG (Arlesheim, Switzerland), CBN (>95%) from CBDepot s.r.o. (Teplice, Czech Republic), and Δ^9^-THC (>95%) from PharmaCan s.r.o. (Prague, Czech Republic). Methanol, acetonitrile, and formic acid were all LC-MS grade and purchased from Merck (Prague, Czech Republic). Flower Gold sunflower oil, with saponification value of 189–195 mg KOH/g and ratio of 12:88 saturated:unsaturated fatty acids, was procured from Eseltix (Bratislava, Slovakia). Ultrapure water used for the LC-MS analyses was obtained in-house from a PureLab Ultra system (Elga, UK).

### 2.2. Stability Study

Sample preparation for stability studies was conducted by weighing 5 ± 0.1 mg of the CBD powder into 2 mL glass vials, which were then placed in the dark within stability chambers and under appropriate conditions without any further modification. In the case of the CBD oil samples, 1 mL of sunflower oil, whose exact mass was determined by weighing (KERN ABP 200-5DM, Balingen, Germany), was added to the CBD powder. An ultrasonic bath (Bandelin Sonorex, Berlin, Germany) at 35 kHz for 3 min was used to completely dissolve CBD in sunflower oil.

All of the samples were prepared in duplicate for each condition (temperature and humidity) and time point in open and closed vials to address the possible influence of free access of air to the CBD samples. Closed vials (screw caps) were wrapped with Parafilm (Pechiney Plastic Packaging, Ohio, USA).

The prepared samples were placed in the stability chambers at Quinta Analytica, s.r.o. (Prague, Czech Republic). According to ICH guidelines [15], the recommended storage conditions for both forms of CBD were 25 °C/60% RH and 40 °C/75% RH for 7 and 14, 30, 60, 90, 180, 270, and 365 days. Photostability tests included the preparation of four samples and their exposition to UV and Vis light, respectively. Those samples were prepared by the same procedure as mentioned above. The required conditions were total exposure in duration of 5 days at least 1.2 million lux hours to the Vis and another 5 days to 200 W hours/m^2^ to the UV light source. The source used to illuminate Vis radiation was equipped with TL D W18/33 640 fluorescent tubes and the UV intensity measurements were performed with a standard UV spectroradiometer in the spectral range of 350–405 nm.

After exposure to stability conditions, the viscosity of samples was measured with a HAAKE RheoStress 600 instrument (Waltham, Thermo Scientific, USA) using a PP35 sensor at a constant temperature of 25 °C.

### 2.3. Preparation of Stability Samples and Reference Standard Samples before LC-MS

At definite time intervals, the stability samples were taken from the stability chambers and stored in a freezer in the dark at −50 °C until LC-MS measurement. Samples for LC-MS analysis were prepared by dilution in methanol to a final concentration of 0.5 mg/mL. The CBD oil samples required a more complex sample preparation protocol. In the first step, the oil samples were diluted 100 times with methanol. Due to the limited miscibility of oil with methanol, the samples were intensively treated in an ultrasonic bath at 35 kHz for 3 min and then further diluted 10 times with methanol. The resulting homogenous solution was injected directly into the LC-MS system. Reference samples which were not exposed to the environmental influences were diluted with methanol and injected in LC-MS as described above. The area under the peak of reference samples was considered as the standard form determining the degradation in the stressed samples.

### 2.4. LC-UV-MS Analysis of Stability Samples

The experiments were performed using a Dionex Ultimate 3000 HPLC system (Waltham, Thermo Scientific, USA) composed of a dual pump, an autosampler, a column thermostat compartment, and a diode array detector (DAD). An Eclipse C18 Plus column (Agilent Technologies, Santa Clara, CA, USA) with dimensions of 50 mm length and 2.1 mm internal diameter with particle size of 5 µm was used for was used for separation. The column was equipped with a guard column with dimensions 5 mm length and 2.1 mm internal diameter, packed with the same sorbent. The column oven and autosampler were thermostated at a temperature of 25 °C. Detection was performed simultaneously with the DAD and a triple quadrupole mass spectrometer (3200 Q TRAP, AB Sciex, Ontario, Canada) with electrospray ionization (ESI) in a positive mode. Analyst 1.6 (AB Sciex) software was used to acquire and evaluate data from the LC-UV-MS system. The mobile phase consisted of ultrapure water containing 0.1% formic acid as phase A and acetonitrile containing 0.1% formic acid as phase B at a flow rate of 0.2 mL/min. The LC gradient had the following time profile: 0.0–10.0 min linearly from 5% to 100% B; 100% B was maintained from 10.0 to 12.0 min; then from 12.0 to 12.5 min 100% to 5% B. Finally, 5% B was maintained to 20.0 min. The injection volume was 5 µL. Serial detection was performed using the DAD, where the UV spectra were scanned in the range of 200–400 nm and MS detection took place with a mass range of 80–1550 *m*/*z*. Quantification of the powder and CBD oil samples was based on area normalization of automatically integrated peaks with appropriate weight correction of samples using UV detection at 225 nm and 210 nm, respectively.

### 2.5. Statistical Analysis

All statistical data presented in the following text and *p* values were calculated by applying Student’s significant test (MS Excel 2010, USA).

## 3. Results

### 3.1. HPLC-MS Analysis

Cannabidiol and its potential degradation products were separated on a C18 column by gradient elution, as described in Section 2.4, utilizing simultaneous UV and MS detection. The addition of formic acid to the mobile phase improved chromatographic resolution and provided better peak shapes, while working at a flow rate of 0.2 mL/min provided good chromatographic performance (Figure 2a,b).

Each sample exposed to specified conditions was measured four times. The test used to detect a possible decrease in the amount of CBD over time was based on a statistical comparison of multiple samples to determine whether the average peak area values of the reference standard sample statistically differed from the average areas of the real stressed samples taken from the stability chambers at a specific time. For this purpose, Student’s *t*-test was applied with a significance level of *α* = 0.05.

### 3.2. Evaluation of CBD Stability Samples

The results of the CBD powder stability study are presented in Figure 3. As can be seen, CBD in the form of powder is stable through almost the whole of the testing period. The integrity of the CBD exposed to 25 °C ± 2 °C/60% RH ± 5% remained statistically unchanged for 270 days. The samples stored in open vials for one year showed a slight decrease of 10.37 ± 0.51% in total (*p* = 9.43 × 10^−6^). The samples stored in closed vials under the same conditions (25 °C ± 2 °C/60% RH ± 5%) also showed statistically significant differences after one year of storage (*p* = 7.42 × 10^−7^) with a smaller decrease of 8.01 ± 0.67% in total.

The stress conditions corresponding to 40 °C ± 2 °C/75% RH ± 5% led to a statistically significant decrease in the amount of CBD in samples stored for 180 days (*p* = 7.91 × 10^−7^) in both open and closed vials. The observed average decrease was 8.21 ± 0.57%.

In addition to temperature and humidity, another important parameter was free versus restricted air access to the samples (open vs. closed vials). The results showed that the change in the amount of CBD stored as solid powder depended mainly on temperature. However, the differences between samples stored in open and closed vials seemed slightly more pronounced (but statistically insignificant) at 25 °C ± 2 °C/60% RH ± 5% after 365 days.

When stored at 25 °C ± 2 °C/60% RH ± 5%, CBD in sunflower oil was stable for a period of at least 180 days. The first statistically significant difference between the samples was observed at 270 days in open vs. closed vials (*p* = 3.17 × 10^−6^), where in open samples, a CBD decrease of 11.41 ± 0.45% compared to samples stored in the closed vials was measured (Figure 4). After a storage period of one year, the amount of CBD reached a value of 58.03% of the original amount, which corresponds to an average total loss of 41.97% of CBD in open and closed vials. Using the *t*-test, calculated *p* value showed no significant difference between the amount of CBD in the samples stored in open and closed vials for the whole testing period.

Unlike the CBD powder samples, the CBD oil samples stored at 40 °C ± 2 °C/75% RH ± 5% in open vials underwent significant degradation as early as after 90 days, with a loss of 20.2% of CBD (Figure 4). Moreover, there was a very significant difference between samples stored in open and closed vials. The analysis of samples stored in closed vials showed no statistical decrease in the amount of CBD after 90 days under the same conditions. After 180 days, however, a significant CBD loss of 16.5% (*p* = 3.68 × 10^−12^) was determined. After a storage period of one year, the amount of CBD stored in the closed vials reached a value of 24.09%, with total degradation of 75.91%, while the remaining amount of CBD in open vials showed a value of 1.03%, corresponding to a total degradation of 98.97% CBD.

All of the oil samples were weighed before and after the storage period. This eliminated any potential imperfections in the CBD amount determination caused by possible oil evaporation, and no oil evaporation was registered.

Our preliminary experiment confirmed that samples of sunflower oil displayed Newtonian behavior, where their viscosity depended on the shear-strain rate. The viscosity of samples stored in open vials at 40 °C ± 2 °C/75% RH ± 5%, increased from 0.045 Pa.s to 3.62 Pa.s after 365 days (Figure 5). Although a statistically significant increase in viscosity was also observed for oil samples in the closed vials after 90 days (*p* = 0.001), the change in viscosity of the samples in the closed vials was much slower. As the viscosity increased, the solubility of the oil samples in methanol decreased, and a longer ultrasonic mixing was necessary for the sample preparation preceding liquid chromatograph-ultraviolet-mass spectrometric (LC-UV-MS) analysis.

The irradiated samples of CBD in both forms (powder and oil) for 5 days were subjected to statistical analysis of CBD content after assessment for possible visible changes in physical properties such as clarity, color, etc. Figure 6 along with the statistical tests shows that light exposure did not lead to a demonstrable degradation of the CBD samples exposed to 25 °C ± 2 °C/60% RH ± 5%. However, a statistically significant decrease of approximately 4.5% was determined for the oil samples stored at 40 °C ± 2 °C/75% RH ± 5% in the open vials (*p* = 0.043).

Despite not being the main aim of the study, we tried to detect degradation products of CBD during the course of the stability study using LC-UV-MS. Two important potential CBD degradation products were available for us in the form of authentic standards, specifically, Δ^9^-THC and cannabinol (CBN) (Figure 1).

Because all of the CBD stability measurements were performed utilizing UV in series with MS detection in the MS scanning mode, it was possible to register the potential presence of Δ^9^-THC and CBN in partially degraded CBD samples. A detailed analysis of all of the obtained LC-UV-MS chromatograms provided evidence of CBN in many partially degraded samples, whereas Δ^9^-THC was not found. However, it should be noted that the detection limits of the quadrupole mass spectrometer used in the scanning mode is rather limited and a possible low amount of Δ9-THC may have been left undetected. In addition, we found several oxidation products of CBD eluted at various retention times in partially degraded samples, as demonstrated by the presence of [M+H]^+^ 331.1 *m*/*z* and 347.3 peaks in the MS chromatograms [12].

## 4. Discussion

An integral part of the systematic approach to evaluating a drug stability is to verify its behavior under the influence of various external factors. The stability assessment, the length of the stability study, and the storage conditions are usually the main focus of a pharmaceutical scientist’s attention in the development of all dosage forms. In general, the drug should be evaluated under storage conditions, with appropriate tolerances, to verify its thermal stability and moisture sensitivity. Stability test design must take in consideration the nature of the substances and geographic area in which the drugs will be used [15].

In order to evaluate the effect of heat, humidity, oxygen access, matrix, and light, it was necessary to find an effective, sensitive, and selective analytical method for the detection and determination of CBD and its possible degradation products. The literature refers to a few studies where cannabinoids were analyzed in various matrices using isocratic and gradient elution profiles, while a combination of UV and MS/MS was used to measure cannabidiol-rich products [16,17,18,19]. Separation of cannabinoids under isocratic conditions is a challenging task due to their physical and chemical properties [17,20]. Therefore, we chose the gradient profile, using acetonitrile and water. All samples were measured by the same LC method.

The results of the CBD powder stability study are in correspondence with a stability experiment performed by Turner et al. [21], who studied the influence of higher temperatures on dried cannabis plant material. They showed that a significant loss of cannabinoids is caused by thermal exposure at 37 °C and 50 °C in the first 10 weeks, while the content of CBD in all of the stored materials remained relatively constant without any detectable decomposition for 100 weeks.

Considering the fact that our experiments were performed in stability chambers, light played no part in the decomposition of CBD in the above-mentioned stability tests. In this regard, the results cannot be in total correspondence with data published by Trofin et al. [22], who showed that the content of CBD in cannabis oil, obtained by extraction from herbal cannabis decreased with storage time after light exposure. They observed a steady decrease of Δ^9^-THC and CBD over the entire storage period of four years, which was more pronounced in samples exposed to light at 22 °C than in samples stored in the dark at 4 °C. The average loss of CBD recorded during each year was 4–5.5%. At the same time, the content of CBN continuously increased during the storage period with an average gain of 4.5%. Nevertheless, it may be concluded that the behavior of CBD does not deviate much depending on the used oil matrix.

In addition to a statistically proven decrease in the amount of CBD after a specific time under given conditions, certain other physical and chemical changes to the oil samples were observed during the course of our stability test. Specifically, a slight change in color to a darker yellow was observed in the case of the samples stored for 180 + days. This was accompanied by an increase in viscosity of samples stored for 180 + days at 40 °C ± 2 °C/75% RH ± 5% (Figure 5) (vide supra). A rapid color development of heated oil mentioned by Maskan may be correlated to the chemical processes taking place in the sunflower oil samples [23]. A change of color of the oil from orange to brown indicates the processes of oxidation, polymerization, and other chemical changes, which also lead to an increase in the viscosity of heated oil [24]. Therefore, the observed change in color and the increase in viscosity mainly for samples exposed to 40 °C ± 2 °C/75% RH ± 5% is likely attributed to the higher temperature and the influence of oxygen in open samples, resulting in the formation of complex structures in the oil [25]. There is an apparent link between the change of oil viscosity after 90 days at higher temperatures and the incipient CBD degradation under such conditions. In their work, Pavlovic et al. [26] compared olive oil, hemp seed oil, and medium chain triglycerides (MCT), all three containing the same amount of CBD. They determined that MCT oil was more stable and more likely to maintain a consistent flavor and visual characteristics over time, which improved the bioavailability of CBD.

Photostability studies, necessary to evaluate the overall photosensitivity of CBD, are very important due to a possible change that may occur when a product is stored under light exposure. According to some authors, cannabinoids are less stable in light, but this also depends on other conditions such as the solvent in which they are stored, as well as temperature, oxygen access, and many other factors. Fairbairn et al. [27] showed that light-exposed samples stored in various solvents degraded faster than samples stored in the same solvents in the dark, while Turner and Hanry [28] reported that THC and CBD either in solution or in crude extract form were stable for six days when exposed to both natural and artificial light. The main conclusion that may be derived is that the exposure to light alone for the given time period does not lead to significant changes in the CBD content, but in combination with other factors, such as the solvent used, increased temperature, and presence of oxygen, light may accelerate the degradation process.

## 5. Conclusions

While CBD stored in the form of a solid powder remained mostly intact over a one-year time period under the studied stability conditions with an approximately 10% decrease in CBD, the influence of temperature, humidity, and air oxygen on CBD oil samples was much more pronounced. Very significant degradation of CBD took place between the 90th and 180th day in the case of CBD oil samples in open vials at 40 °C ± 2 °C/75% RH ± 5%, with a complete degradation of CBD after approximately 270 days. Exposure to artificial light showed that the light itself does not have strong adverse effects on CBD. Nevertheless, in combination with other factors, it may accelerate the degradation process.

The experimental results of our CBD stability study showed that under defined stability conditions, various degradation products, including CBN and several oxidation products, were formed. However, Δ^9^-THC was not detected by our method.

## Figures and Tables

**Figure 1 pharmaceutics-13-00412-f001:**
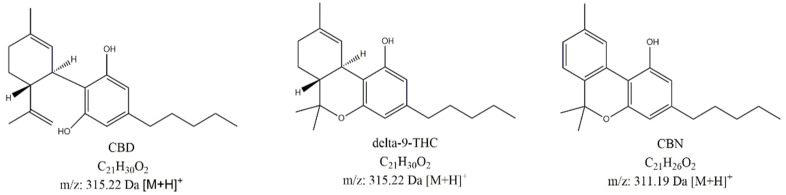
Chemical structures of active ingredient cannabidiol (CBD) and its possible degradation products (Δ^9^-tetrahydrocannabinol (Δ^9^-THC) and cannabinol (CBN)).

**Figure 2 pharmaceutics-13-00412-f002:**
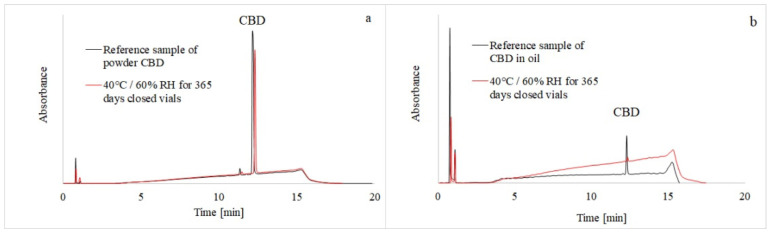
HPLC-UVchromatograms of CBD reference samples (black line) and CBD samples stored for 365 days (red line) in the stability chambers. The CBD powder was measured at 225 nm (**a**) and the CBD oil solution at 210 nm (**b**).

**Figure 3 pharmaceutics-13-00412-f003:**
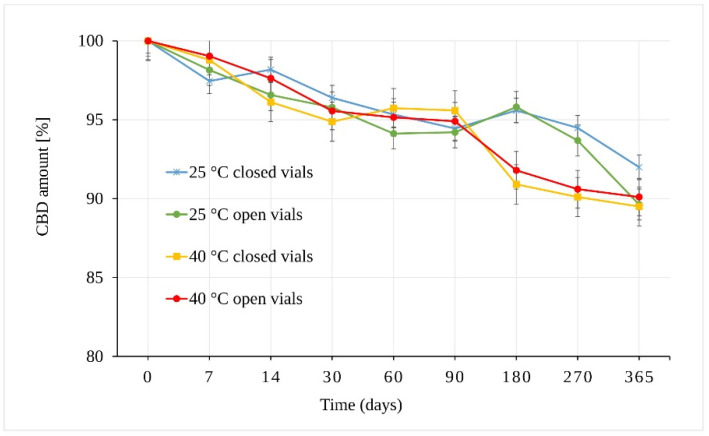
Stability of the CBD powder samples expressed as the amount of CBD over time measured by LC-UV (225 nm).

**Figure 4 pharmaceutics-13-00412-f004:**
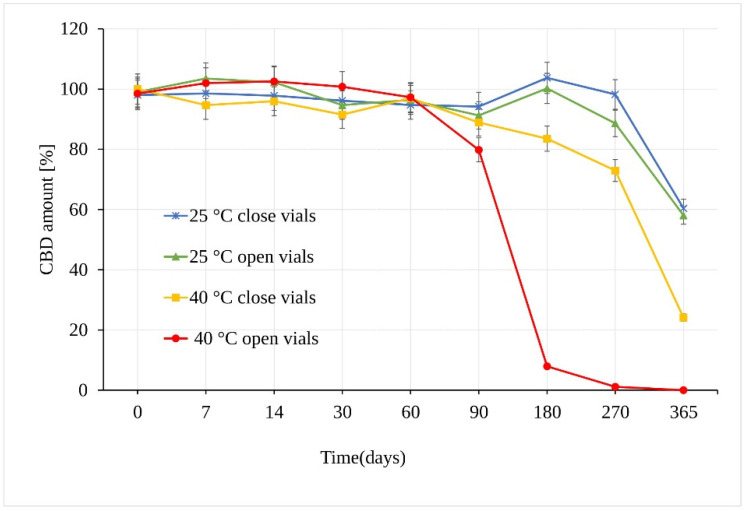
Stability of the CBD oil samples expressed as the amount of CBD over time measured by LC-UV (UV 210 nm).

**Figure 5 pharmaceutics-13-00412-f005:**
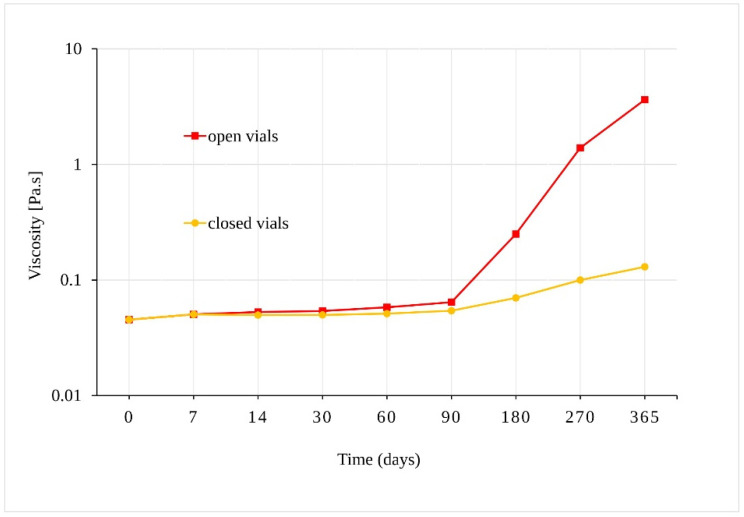
Dependence of the viscosity of the oil samples stored at 40 °C ± 2 °C/75% RH ± 5% in closed and open vials.

**Figure 6 pharmaceutics-13-00412-f006:**
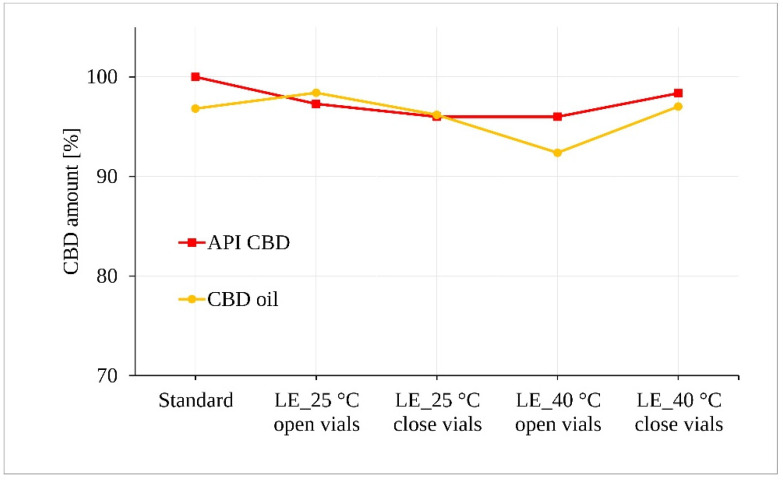
Photostability of the CBD samples measured by LC-UV. (LE, light exposure).

## Data Availability

The source data available upon a reasonable request.

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
