# Peer review of "Stability Study of Cannabidiol in the Form of Solid Powder and Sunflower Oil Solution"

_pharmaceutics, 2021, doi:10.3390/pharmaceutics13030412_

Round 1

Reviewer 1 Report

 A degradation study of cannabidiol by LC-UV-MS

 Ema Kosović 1, 2, David Sýkora 2 and Martin KuchaÅ™

Comments:

  1. The title mentions degradation study : but the abstract and the contents does not mention if the authors studied degradation products and mechanism of degradation, which makes the title less appropriate to justify the work. A more relevant title would be “Stability of cannabidiol in solid powder and oil vehicle under accelerated and long term stability conditions.”
  2. Abstract: Line number 13: the sentence can be rephrased to “the aim of the present study was to investigate the chemical stability of CBD in the form of solid API powder (herein referred to as CBD powder) and also when formulated as oily solution”. It is not clear if CBD was dissolved in an oil, when the authors mention “oily solution. In that case, the authors are recommended to mention the name of oil that was used.
  3. Abstract : The line number 15 indicates an old version for the full form of ICH. The latest version of ICH is to be updated in the full-form. Line number 16 indicates single doses of CBD. It is not clear what is a single dose? It should be indicated if the authors used marketed formulation or they isolated and extracted the CBD. In line 17, the authors can mention that CBD powder and oily solution were exposed for 7,14, 30, 60, 90, 180, 270 and 360 days rather than the usage of term “months” at two different stability conditions namely, 25°C/60 %RH and , 40°C/75 %RH. It is important to mention that both the solid drug and oily solution were exposed to the long term storage and short term storage conditions at all the time points.
  4. In the abstract: line numbers 19-24, the authors state something very obvious. Degradation in solid state (powder) would be lesser than the one in oil-solution (this does not make an innovative conclusion). It would be useful in that case to justify the therapeutic significance of stability of oily solution in pharmaceutical formulation.
  5. In the Introduction: line number 32: The figure, 1 also mentions abbreviation CBN which is cannabinol. The full form along with the abbreviation should be listed in line 32.
  6. Line 37: authors should rephrase as CBD is the second most.. of cannabis and it does not have any psychoactive effects. Introduction covering points until lines 57 are extensively detailed. The authors are requested to make it precise and stress more on the stability aspects citing the appropriate works and their results.
  7. Line 66: authors can replace the terms “pharmaceutically common CBD forms” with “commercially marketed formulations of CBD”.
  8. Line 67 : authors should rephrase the sentence as : the stability of CBD was evaluated in the form of a solid powder and an oil solution for one year duration, under ICH stability conditions.
  9. Line 69: “the study was adapted to climate conditions …” the sentence does not sound relevant. Experimental conditions are set in stability chamber and are defined by the ICH guidelines and do not need the mention of the country where it was carried out.
  10. Line 71: abbreviation of UV is missing in the full-form. The full form should be stated as “ultraviolet-visible (UV-Vis) spectroscopic…”
  11. Figure 1. The authors need to state that CBD is the active ingredient and Δ9-THC, CBN are the reported degradation products.
  12. The section 4. Materials and Methods has to come as second section, succeeding the first Introduction section.
  13. Lines 248-249 should be scripted as : Methanol, acetonitrile, and formic acid were all LC-MS grade and purchased from Merck (Czech Republic. Flower gold sunflower oil was procured from Eseltix (Slovakia). The authors did not state the quality of the oil. Appropriate measures like Fat index and saponification value can be a predictor of the initial quality of the product.
  14. Line 253: the term “required” has to be replaced with “were conducted by”. The authors must have used a high precision analytical balance to weigh accurately about 5 mg of CBD. The name of precision balance is to be stated.
  15. Line 254: the authors are using a term “stability boxes” which is not understood. A more appropriate term would be “stability chambers”. This replacement has to be made elsewhere in the contents as well.
  16. Lines 257-258: the authors must state how much time it took to dissolve the samples under ultrasonic bath and at what frequency (Hertz) was it operated.
  17. If the drug was known to be photosensitive as indicated in the introduction, then the authors should have used an amber coloured glass vial. Line 254 presumes that the used vials were not amber coloured and that there could be a possibility of light exposure inside the stability chamber.
  18. Line 265: authors should mention the exposure durations in single units of days to be more exact and uniform elsewhere in the contents as well.
  19. Line 266: the authors are not stating how to prepare four samples for exposure to photo stability conditions and which were the said samples. A need to mention!
  20. Line 272: the authors should rewrite the sentence as: After exposure to stability conditions the viscosity of samples were measured with….
  21. Line 276: replace the term “given” with “definite”.
  22. Line 277: must be rephrased as: Samples for LC-MS measurements were prepared by dilution in methanol to a final concentration of 0.5 mg/ml.
  23. Lines 280-281 : it is not clear why the authors did not use acetonitrile as the diluent instead of methanol, as the mobile phase component in LC-MS is already acetonitrile. Authors should indicate if there was a solubility issue with the use of acetonitrile. Line 282: the authors should indicate the time and frequency of ultra-sonication required for complete dissolution of the oily sample.
  24. In section 4.2. it is not clearly mentioned as to whether the authors placed any respective controls along with the CBD powder and oily samples. Control samples would consist of sunflower oil along with samples; CBD powder and CBD powder in sunflower oil.
  25. Lines 284-285: should be rephrased by: reference samples which were not exposed to stability conditions were diluted with methanol and injected in LC-MS as described above. The area under the peak of reference samples was considered as the standard, for determining the degradation in the stressed samples.
  26. Line 287 : title of the section to be re written as : LC-UV-MS analysis of stability samples.
  27. Line 290-291 to be rewritten as : “An Eclipse C-18 Plus column (Agilent Technologies, USA) with dimensions of 50 mm length and 2.1 mm internal diameter with particle size of 5um was used for separation of samples.” The authors are also recommended to mention the dimensions and the brand of guard column used for measurements. In this section the authors have not mentioned the column oven temperature and auto -ampler temperature used for measurements and has to be mentioned.
  28. Line 293: It is not mentioned as to why the authors did not perform measurements in negative ESI mode. Altogether it is not clear as to why was MS measurement performed. The authors did not indicate the elucidation of a degradation pathway..
  29. Line 295 : replace “control” with “evaluate
  30. Line 302: the authors did not mention how the quantification was performed. They need to be specific of the technique used (based on the description it seems an Area normalisation method was used with appropriate weight of samples). Similarly in line 303, the authors do not mention the basis of selecting two different wavelengths although they analyse the same component (CBD) in each formulation. Moreover the authors need to mention the selected integration technique. It is preferred to that the integration be automatic to reduce the operator bias in evaluating the peak’s area.
  31. Line 305: were calculated by.
  32. In the section 2. Results the authors are indicating chromatogram without any overlay of the exposed stability samples. This is rather concealing and an overlaid chromatogram of the maximum exposure time of samples (one year in this case) should be atleast shown as a representative to demonstrate the efficacy of the method to separate and resolve the degradation products from the CBD peak. Additionally authors can add photographs or images to justify the visual appearances after stability exposure. The masses of observed degradants after stability exposure would need a justification with appropriate structure elucidation pathway to support the hypothesis that CBD degrade under the influence of temperature, humidity and oxidation more justifiable.
  33. Line 96 : title of section to be written as : “Evaluation of CBD stability samples
  34. Line 100: 9th month to be replaced by “9 months”or more appropriately “270 days”. Similar corrections should be made elsewhere in the paper.
  35. The authors have to justify in Figure 3 as to why the CBD amount (%) increases from 7 days to 14 days- (blue line) and from 3 months to 6 months (Blue and green lines). This is counterintuitive. Interestingly the degradation difference between the samples stored in open and stored conditions is more pronounced at 25°C/60 %RH conditions rather than 40°C/75% RH. How do the authors justify the differences?
  36. Line 158: the authors have to justify how a degradation of 4.5% can be called slight statistically significant?
  37. Lines 167-169: the authors do not mention the details of the procured reference standards in the materials section (confirmed purity and potency details).
  38. Line 180: Discussion: Although the authors describe and debate on the available literature as to what all factors might lead to degradation of CBD, there isn’t an evidence to be specific as to which factor is the significant contributor to degradation. It could be temperature, oxygen (oxidation) or even humidity in the present case. Hence, the authors are requested to make deductions in this regard more appropriate by analysis of LC-MS spectra and provide some information as to which factor is the most contributing in the present study. The use of elevated temperatures and humidity upon long term exposure (1 year) makes a significant difference in both the powder and oily forms. However, the effect of oxidation in the solid state and more importantly in the oily solution state has a huge significance. The oily vehicle (sunflower oil) can undergo autoxidation and lead to the generation of autoxidation products that may in turn catalyse the degradation in liquid state. There isn’t a mention of its quality or analytical purity of the used oil, anywhere in the contents nor is a mention if the oil solution was kept alone as a control during the study. If available, that data could support some hypothesis for degradation by oxidation alone. Here a significance of the oily liquid in formulation and drug delivery (injectables) can be ascertained by citing appropriate literature. Overall the present study needs to be made novel and more specific to set the foundation for appropriate control of instability factors. This also warrants the use of adequately protective packaging design. If the drug is sensitive to moisture, an impervious packaging should be made while conducting the study. If the drug is sensitive to degrade by oxidation, a packaging that avoids oxygen atmosphere (nitrogen filled) needs to be used. Alternatively the study could be conducted in the vials (flushed in headspace with nitrogen cover) to check the effect of thermal degradation alone. A comparative study with that of a finished pharmaceutical product could be made within the packaging or container closure system used by the manufacturer as is the recommendation by ICH.
  39. Line 323: Conclusions: the selected conditions are not stress conditions. They have to be replaced as the “accelerated stability conditions”. The authors have concluded the possibility that oxidation might accelerate degradation, but not conducted a study where the sample could be protected from external oxygen (such as by using a nitrogen blanket over sample vials or flushing the sample headspace in a desiccator with nitrogen and placing the samples openly inside the desiccator.). The conclusion should specifically highlight the significance of degradation in oily solutions and formulations.

Author Response

Dear respected reviewer,

thank you for your valuable suggestions and comments. Please see the attachment.

Reviewer 2 Report

In this article, a degradation study of cannabidiol in powder and in oil has been evaluated under different temperature and humidity conditions.

I would like to be clarified on these points:

  • In all the experiments you chose to have only 2 repetiotions. Why not triplicates?
  • Why the samples were frozen at -50 C instead of analyzing them directly? This it would have given more reliability to your measurements.
  • Why you didn’t test the samples also not in dark environment?
  • In case of the viscosity measurements, it would have helped a lot to have oil without CBD to see if the viscosity changes due to oil’s properties and not from the presence of CBD. Why you dodn't include this control samples?

Author Response

Dear respected reviewer,

thank you fro valuable suggestions and comments.  Please see the attachment.

Reviewer 3 Report

Dear Editor,

Thank you for the opportunity to review this interesting paper. Although the Authors performed very important analytical studies, the paper should not be accepted in the present form. Please find below my remarks:

  1. The representative chromatograms obtained at the end of the study should be shown. All observed degradation products should be mentioned and based on MS spectra, and their structures should be proposed.
  2. According Q1A(2R) guideline if long-term studies  are  conducted  at  25°C  ±  2°C/60%  RH  ±  5%  RH  and    “significant  change”  occurs  at  any  time  during  6  months’  testing  at  the  accelerated  storage  condition,  additional  testing  at  the  intermediate  storage  condition  should  be  conducted  and  evaluated  against  significant  change  criteria.  Testing  at  the  intermediate  storage  condition  should  include all tests. The intermediate storage condition is: 30°C ± 2°C/65% RH ± 5% RH during 6 months. Based on presented in this paper results, it can be concluded that “significant changes” were observed for CBD oil samples.
  3. The viscosity properties of the CBD oil samples should be compared with the results for oil samplses without CBD (blank samples) and statistical analysis should be performed.
  4. The conclusions should be more condensed (briefly written) and present unambiguous summaries of research results

Author Response

Dear respected reviewer,

thank you for valuable suggestions and comments.  Please see the attachment.

Round 2

Reviewer 1 Report

Acceptable, all points addressed.

Reviewer 3 Report

The Authors improved some apsects and the paper can be consider fort publication.